# The Differences in Histoarchitecture of Hoof Lamellae between Obese and Lean Draft Horses

**DOI:** 10.3390/ani12141774

**Published:** 2022-07-11

**Authors:** Magdalena Senderska-Płonowska, Natalia Siwińska, Agnieszka Zak-Bochenek, Marta Rykała, Malwina Słowikowska, Jan P. Madej, Katarzyna Kaleta-Kuratewicz, Artur Niedźwiedź

**Affiliations:** 1Department of Immunology, Pathophysiology and Veterinary Preventive Medicine, Wroclaw University of Environmental and Life Sciences, 50-137 Wroclaw, Poland; agnieszka.zak-bochenek@upwr.edu.pl (A.Z.-B.); jan.madej@upwr.edu.pl (J.P.M.); 2Department of Internal Medicine and Clinic of Diseases of Horses, Dogs and Cats, Faculty of Veterinary Medicine, Wroclaw University of Environmental and Life Sciences, 50-366 Wroclaw, Poland; natalia.siwinska@upwr.edu.pl (N.S.); marta.rykala@upwr.edu.pl (M.R.); malwina.slowikowska@upwr.edu.pl (M.S.); artur.niedzwiedz@upwr.edu.pl (A.N.); 3Department of Biostructure and Animal Physiology, Faculty of Veterinary Medicine, Wroclaw University of Environmental and Life Sciences, 50-375 Wroclaw, Poland; katarzyna.kaleta-kuratewicz@upwr.edu.pl

**Keywords:** obesity, laminitis, horses, equine metabolic syndrome, draft horses

## Abstract

**Simple Summary:**

Modern research indicates a significant role of obesity in equine health. Obesity is associated with equine metabolic syndrome (EMS), which affects hoof lamellae, causing a painful condition known as laminitis. However, it is not known if obesity itself can cause lamellar failure. Forelimbs from 12 draft horses (six obese and six lean) were acquired. To exclude EMS animals, the insulin concentration was measured in the blood. The hooves were radiographed and assessed histologically. Lamellae differed between obese and lean animals; the damage was found in both groups. However, it is hard to say which group had more laminitic changes: the present study revealed that, in the obese group, 89% of the primary dermal lamella (PDL) was standard, but in the lean group the percentage was only 58%. Therefore, the study did not support the effects of obesity on lamellar failure. The measurements taken indicate that the lamellae are much longer compared to other research studies; this could indicate that the length of the primary epidermal lamellae (PEL) depends on the hoof size. Regardless of previous results, this is the first research showing the anatomy of lamellae of draft horses.

**Abstract:**

Obesity is a common problem in horses. The associations between obesity and equine metabolic syndrome (EMS) and between EMS and laminitis are known. However, there is a lack of data on whether obesity itself can affect hoof lamellae. Forelimbs and blood from 12 draft horses (six obese and six lean) from a slaughterhouse were acquired. To exclude laminitis and EMS horses, insulin concentration was measured, and hooves were radiographed. Histological evaluation was performed. The shape of the primary and secondary epidermal lamellae (PEL and SEL) was evaluated, and the length of the keratinized and total primary epidermal lamellae was measured (KPEL and TEL). All horses showed pathological changes in lamellae. In the lean group, the changes were longer SELs, more proliferated and separated PDLs, and less standard PDLs. In the obese group, the changes were a lower number of club-shaped and standard SELs and significantly more tapered SELs. No difference in the shape of PELs and the length of KPELs was noticed. The research did not confirm the effects of obesity on lamellar failure. The measurements taken indicate that the lamellae are much longer compared to other research studies; this could indicate that the length of the PEL depends on the hoof size.

## 1. Introduction

Equine obesity is an increasingly common problem in highly developed countries, especially in ponies [1,2,3,4,5,6]. It can be an issue in itself, but it can also be a predisposing factor to other medical conditions such as systemic inflammation, hyperlipemia, osteochondrosis, or equine metabolic syndrome [7,8]. The latter is under particular scrutiny because of its similarity with human insulin resistance. Equine metabolic syndrome is a condition in which the animal’s body develops insulin dysregulation [9]. This condition especially affects ponies but also warmblood breeds such as Arabian, Spanish, or Morgan [10]. The age range was not stated; however, a recent study in the UK shows that the median age for diagnosis of EMS was 11 years old with an interquartile range between 9 and 13 [6]. There is an ongoing debate over whether obesity in humans is caused by insulin dysregulation or vice versa [11]. Understanding insulin dysregulation is crucial for equine practitioners because it is one of the main factors leading to laminitis, a life-threatening and common illness among the equine population [12]. 

The metabolic path was not confirmed until 2007. It was found that intravenous administration of insulin to healthy ponies results in laminitis in all examined animals [13]. Three years later, this condition was also confirmed in warmblood horses [14]. The pathophysiology of lamellar failure due to hyperinsulinemia is unclear [15]. Although there are many reasons for developing laminitis, the most common cause in highly developed countries is endocrinopathic issues: 89% of horses with laminitis admitted to hospital in Finland suffered from this condition [16]. Endocrinopathic laminitis develops very slowly, and the affected horses stay in subclinical laminitis for a long period before the acute phase [17].

The condition of lamella can be evaluated by histological examination. During laminitis, morphological changes appear in the stratum internum, also called the inner hoof wall. The stratum internum consists of 550–600 primary epidermal lamellae (PELs) whose outer ends reach the keratinized stratum medium. To increase the surface area for the attachment of the epidermis with the deeper located dermis, each primary epidermal lamella divides on the periphery into many small protrusions called secondary epidermal lamellae (SELs). Several histological studies have found typical changes for lamellar failure concerning sepsis-related laminitis [18,19] and endocrinopathic laminitis [13,14,20]. The changes include elongation of the PEL, keratinized primary lamellae (KPELL), and total length of epidermal lamellae, effecting the shapes of the axial tip of the PEL and the abaxial tip of the PEL, called the primary dermal lamellae (PDL), and the SEL [14].

Draft breeds do not belong to the group of horses with high EMS risk. What is more, the choice of these horses was practical: it was easy to find horses from one owner with the same age. 

Obesity and insulin dysregulation are closely related, so it was important to conduct the research on the obesity itself, as it can affect lamellar failure. The aim of this study was to evaluate histological changes in lamellae between obese and lean draft horses.

## 2. Materials and Methods

### 2.1. Material

The front legs below the carpal joint were collected immediately after slaughter from 12 draft horses (one front leg, from one horse; randomly left or right): 6 horses with normal body condition score according to Henneke et al., (1983) [21] (BCS 4-5/9), called “lean”, and 6 extremely fat horses (BCS 9/9) called the “obese group”. Body condition scoring was performed by NS. All the horses were derived from a single slaughtering horse breeder and were the same breed and the same age (5 years).

Blood from the external jugular vein was drawn prior to slaughter to obtain basal serum insulin concentration. All animals were tested at rest (no fasting). Before the test, all horses had constant access to hay. The assessment of insulin activity was performed using IMMULITE 2000 (Siemens, Munich, Germany) in the veterinary laboratory in VetLab Poland. The horses with insulin concentrations above 20 µIU/mLwere excluded because of high risk of EMS [22].

### 2.2. Sample Collection

The forelimbs were disarticulated in the abattoir, and the metacarpophalangeal joints were frozen within 1 h of death. After defrosting, to exclude any changes associated with chronic laminitis, the acquired hooves were radiographed in latero-medial projection, and the rotation was measured. Horses with rotations above 4° were excluded. Then, the hooves were cut with an oscillating saw in a sagittal plane, and the following cut was performed 2 cm laterally from sagittal cut; next, 2 horizontal perpendicular cuts were made in the middle of hoof height at a distance of 2 cm from each other. The resulting cube was finally cut by a scalpel blade to remove the dense (often pigmented) hoof wall and the coffin bone. Each section was divided into 5 mm wide segments, and the resulting samples were fixed in 4% formaldehyde.

### 2.3. Histological and Morphometrical Evaluation

All samples of the hooves were routinely embedded in paraffin wax and then cut into 5 µm sections and stained using Delafield’s hematoxylin and eosin (HE). The samples were examined using a light stereoscope Zeiss Stemi 508 doc (Zeiss, Jena, Germany) with a video camera. The morphometry was performed on microphotographs using ZEN 2.6 software (Zeiss, Jena, Germany). Measurements included total epidermal lamellar length (TELL, µm), keratinized primary epidermal lamellar length (KPELL, µm), and primary epidermal lamellae length (PELL, µm) according to van Eps and Pollitt (2009). Measurements were performed on ten randomly chosen PELs from each section. Examination of the sample includes the classification of PEL and SEL according to Karikoski et al. (2015). Ten adjacent PELs on each sample were evaluated, classified, and measured. Similarly, ten randomly selected SELs were chosen (5 from the left and 5 from the right side) in three different (axial, middle, and abaxial) regions of each previously selected PEL (30 measurements × 10 PELs).

### 2.4. Statistical Analysis

The normality of the obtained results was tested with Shapiro–Wilk test. The differences in insulin concentration—PELL and TELL—between the groups were evaluated with Student’s *t*-test using STATISTICA 13.3 software. As KPELL distribution was not normally distributed, the difference was measured with Mann–Whitney U test. All measurements in lean horses and all measurements in obese horses were combined together. The results show mean and standard deviation (±). A value of *p* < 0.05 was considered significant. For differences between groups in the shape of PDL, PEL, and SEL, the chi-squared test was performed. If the number of comparing groups is <5, the Fisher’s exact test would then be used.

## 3. Results

Insulin concentration was 8.5 µIU/mL± 9.4 for the obese and 5.8 µIU/mL± 9.4 for the lean, no statistical differences were noted (*p* = 0.922), and no horse had insulin levels above 20 µg.

In the radiograph examination, all horses had a rotation less than 4°.

The length of the PEL in the lean group was significantly higher (3.83 ± 0.60 µm (mean ± standard deviation)), than that in the obese group (3.31 ± 0.41 µm) (*p* < 0.001). The TELL for lean was significantly higher (4.29 ± 0.62 µm) than in the obese group (3.80 ± 0.46) (*p* < 0.001). There were no differences between KPELL in the lean group (0.47 ± 0.38 µm) and in the obese group (0.50 ± 0.24 µm) (*p* = 0.653) (Figure 1).

There were no differences between the shape of PELs in the lean and obese groups. Most PELs in both groups had a tapered shape (Table 1). Shapes of the PEL are shown in Figure 2.

The significant differences between the shape of PDLs were noted between the groups. In the obese group, almost all PDLs were standard. Although, in the lean group, the standard shape was also dominating, 38% of PDLs was in a different shape than standard. The most common was proliferative and separated. Of importance could be the fact that the proliferative shape was found only in two specimens, and the separated shape was found in one specimen. Shapes of PDLs are shown in Table 2 and in Figure 3. 

The examined SELs showed the entire spectrum of shapes. There were statistical differences between the groups of obese and lean horses, and these are shown in Table 3 and Figure 4.

## 4. Discussion

This is the first research comparing lamellae between obese and lean horses. It is also the first research evaluating draft horses’ lamellae.

Previous studies indicated that in horses with laminitis, an increase of KPELL caused an increase in the total length of the lamellae (TELL) (van Eps and Pollit, 2009). In our study, KPELL did not differ between groups, as opposed to PEL length. Other authors indicated that in endocrinopathic laminitis, the PEL is slightly longer [20]; however, in mechanical laminitis, the length of PEL does not differ [23]. The length of PEL was higher in lean horses compared to obese horses, and as a result the TELL was also much higher in the group of lean horses. PEL length can be affected by age in horses but not in ponies [20] and by the origin of the sample (distal or proximal, toe or quarter) [22,23,24,25,26], but in our study horses were the same age and only proximal samples of the toes were used. Therefore, we have not yet found an explanation for this finding. Particularly in lean horses, the means of TELL and PELL were much higher in our research (TELL = 4.29 µm, PELL = 3.83 µm) when compared to other studies, where the TELL = 3.23 mm and PELL ~2.8 mm (data read from the graph) (van Eps and Pollitt, 2009) and TELL = 2.55mm [24]. These discrepancies are probably caused by the differences between breeds. In our study, all individuals were draft horses and had much bigger hooves than the warmblood horses or ponies previously examined by the cited authors. 

In the current study, the most common type of PEL was tapered; in healthy thoroughbreds, it is the most common type [22]. However, in healthy ponies and horses, the most common type of PEL is standard [20]. In the study of Karikoski (2015), the tapered shaped of PEL was associated with endocrinopathic laminitis. In studies with experimentally induced laminitis, tapering and elongation of PELs and SELs were typical for laminitis induced by both oligofructose overdose (de Laat, 2011; van Eps and Pollitt, 2010) and hyperinsulinemia [13,14]. On the other hand, the mentioned changes were also found in four out of six three-year-old healthy quarter horses [26]. It was also indicated that mechanical stress may affect the density and curvature of a PEL or its bifurcation [27]. As the definition of laminitis is still not clear [28], it is impossible to claim if the recognized changes are laminitis, breed-associated, or even mechanical for that matter.

It was previously indicated that in laminitic horses only 12% of the PDL was standard while in healthy horses and ponies it was in the range of 74–96% [21]. The present study revealed that, in the obese group, 89% of PDLs were standard, but in the lean group the rate was only 58%. Moreover, in the lean group, some PDLs were proliferative and separated, which was recognized as laminitic changes. Proliferative lamellae could also be a result of healing, for example, after keratoma removal [9]. It is possible that both lean and obese groups could suffer from subclinical laminitis. It was shown that the most severe changes in the course of laminitis are associated with the abaxial region of lamellae (tip of PDL) [20]. The lack of differences between groups in the axial region (tip of PEL) may indicate subclinical laminitis.

The shape of SEL varied between studied groups. Karikoski et al. [21] revealed that, in horses and ponies with endocrinopathic laminitis, the most common type of SEL was tapered (axial), tapered and fused (middle), and keratinized (abaxial). In the current study, the most common type of abnormal SEL was tapered in both groups but more so in the obese group. According to Kawasako (2009) [29], the tapered shape of SEL is common in breeding mares, racing horses, and two-year-old horses but not in foals and yearlings. However, in the research of Karikoski et al. [21], as mentioned before, it is more typical for laminitis cases. In previous studies, the separated shape of SEL was found only in laminitic horses but not in healthy horses [20], while in our results, separated SEL was present in both groups and in every studied region, but it was more pronounced in the obese group. Another morphological feature that characterizes the SEL is a club shape. The results drawn from previous studies according to this parameter are inconsistent. In the study of Karikoski et al. [20], club-shaped SEL was less often found in laminitic (0.3–1.1%) than in normal (0–7.8%) animals. Furthermore, in the studies of Kawasako (2009) [29], in healthy horses, 25–50% of SELs were club-shaped. Our research indicated that, in the lean group, more horses had club-shaped SELs than in the obese group. It is difficult to say whether the club shape has a negative impact on the functioning of the hoof, because in healthy horses it is observed with considerable frequency.

In both groups, we found abnormal lamellae. Signs of laminitis in the lean group could be longer SELs, more proliferated and separated PDLs, and less standard PDLs. In the obese group, there was a lower number of club-shaped and standard SELs, as well as much more tapered SELs. Surprisingly, no difference in the shape of PELs and the length of KPELs was noticed between groups. In general, the lamellae of obese horses seem to be healthier than in the lean group, which is hard to explain. The purpose of these horses may provide a certain explanation; the examined horses were meat horses. We can therefore assume that they were fed more intensively and had less movement than sports or leisure horses. This also could mean that those horses which were not obese could have gone through the disease (e.g., history of pneumonia, colic, diarrhea), restraining weight gain and causing changes in lamellae. Because of the lack of data about the anatomy of lamellae in draft horses, we cannot exclude nor confirm that obtained results are physiological or laminitic. What is more, we cannot exclude insulin dysregulation in the examined horses, because it was impossible to perform a dynamic insulin test. The last things that can affect the result are the animals’ death and freezing the hooves prior to cutting and examination; the exact effect of freezing and death on hoof tissue is as yet unknown.

## 5. Conclusions

According to the authors’ knowledge, this is the first work on draft horses’ lamellae. The findings from this study suggest breed differences in the morphology and length of lamellae may exist, with draft horses appearing to have longer PEL than previously reported in other breeds. As laminitis-like changes were observed in both groups, they are not symptoms directly related to obesity.

## Figures and Tables

**Figure 1 animals-12-01774-f001:**
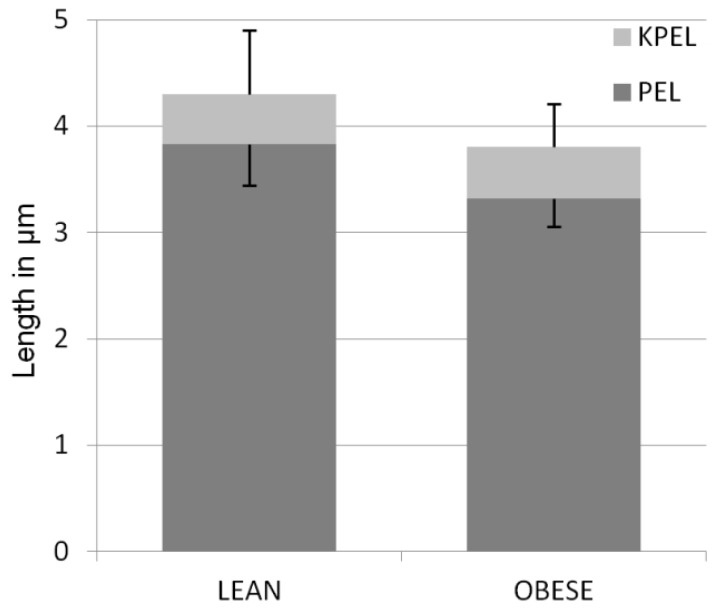
Mean length (±SD) of primary epidermal lamellae (PEL) and keratinized epidermal lamellae (KPEL) in the hooves of lean and obese horses. The difference in PEL length between groups is significant (*p* < 0.001).

**Figure 2 animals-12-01774-f002:**
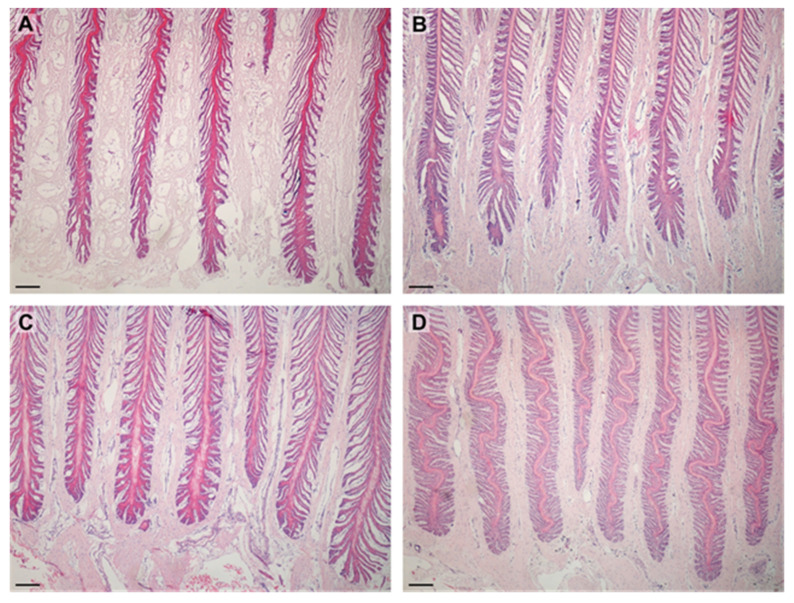
Images show PELs in examined horses. PELs did not differ between groups: (**A**)—sharp PELs in obese horse; (**B**)—sharp PELs in lean horse; (**C**)—standard PELs in obese horse; (**D**)—standard PELs in lean horse. Scale bar = 200 μm.

**Figure 3 animals-12-01774-f003:**
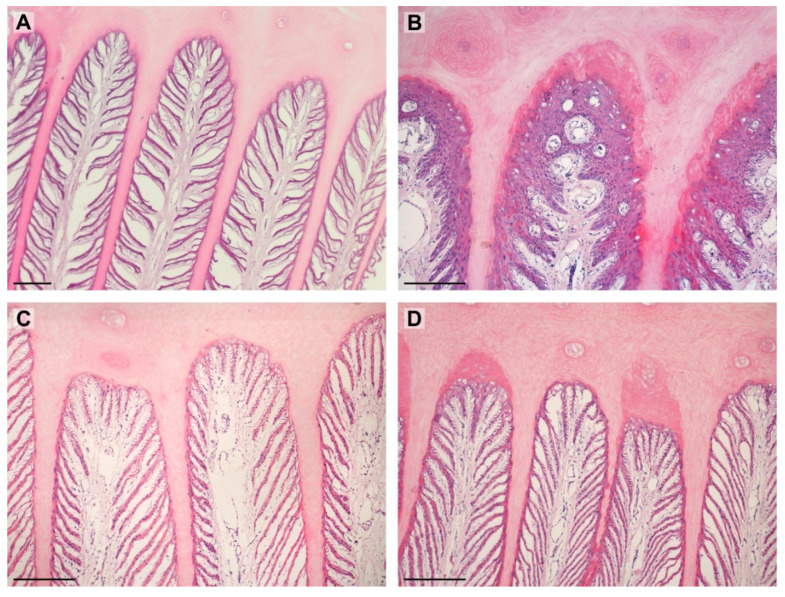
Images show PDLs in examined horses. (**A**)—separated (lean); (**B**)—proliferative (lean); (**C**)—standard (obese); (**D**)—keratinized (obese). Scale bar = 200 μm.

**Figure 4 animals-12-01774-f004:**
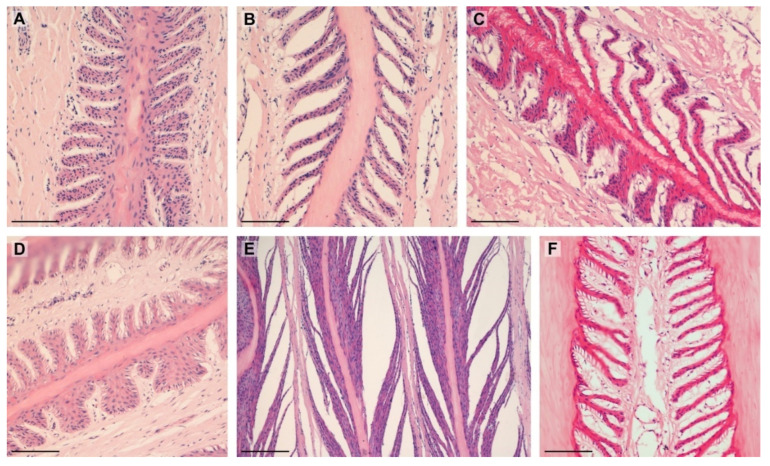
Types of SELs noted in the study: (**A**)—suprabasal layer hyperplasia (lean); (**B**)—tapered (lean); (**C**)—tapering and suprabasal layer hyperplasia (obese); (**D**)—fused and standard (lean); (**E**)—separated (lean); (**F**)—tapered (obese). Scale bar = 100 μm.

**Table 1 animals-12-01774-t001:** Shapes of PELs.

Shape of PEL	Groups		*p*
	Lean	Obese	
Standard	15% (9)	5.4% (3)	0.088
Tapered	58.3% (35)	60.7% (34)	0.794
Sharp	26.7% (16)	33.9% (19)	0.394
Bifurcated ^1^	0% (0)	1.8% (1)	0.482

^1^ Since bifurcation could happen in standard, tapered, or sharp shape, it is shown separately. The number of lamellae is shown in brackets.

**Table 2 animals-12-01774-t002:** Shapes of PDLs.

Shape of PDL (*abaxial PEL*)	Groups		
	Lean	Obese	*p*
Standard	**58.3% (35)**	**89.5% (52)**	**<0.001**
Sharp	1.7% (1)	0% (0)	0.522
Proliferative	**20% (12)**	**0% (0)**	**0.0002**
Separated	**16.7% (10)**	**0% (0)**	**0.0008**
Keratinized	3.3% (2)	8.8% (5)	0.215
Bifurcated	3.3% (2)	1.7% (1)	0.589

The number of lamellae is shown in brackets. Significant differences (*p* < 0.05) were bolded.

**Table 3 animals-12-01774-t003:** Shapes of SELs in examined groups of horses.

Shape of SEL	Groups/localization					
	Lean			Obese		
	Axial	Middle	Abaxial	Axial	Middle	Abaxial
Standard	**16.7% (10)**	**8.3% (5)**	3.3% (2)	**1.7% (1)**	**0% (0)**	0% (0)
*p*	**0.0044**	**0.028**	0.247			
Tapered	**23.3% (14)**	41.7% (25)	56.6% (34)	**40% (24)**	58.3% (35)	65% (39)
*p*	**0.049**	0.067	0.349			
Club-shaped	11.7% (7)	**18.3% (11)**	0% (0)	5% (3)	**0% (0)**	0% (0)
	0.186	**0.0003**	-			
Suprabasal layer hyperplasia	16.7% (10)	10% (6)	**20% (12)**	13.2% (8)	3.3% (2)	**0% (0)**
*p*	0.609	0.143	0.0001			
Fused	13.2% (8)	5% (3)	**0% (0)**	21.7% (13)	5% (3)	**8.3% (5)**
*p*	0.229	1.000	**0.028**			
Separated	16.7% (10)	**16.7% (10)**	**16.7% (10)**	16.7% (10)	**33.4% (20)**	25% (15)
*p*	1.000	**0.035**	0.122			
Keratinized	0% (0)	0% (0)	1.7% (1)	0% (0)	0% (0)	0% (0)
*p*	-	-	0.504			
Bifurcated	1.7% (1)	0% (0)	1.7% (1)	1.7% (1)	0% (0)	1.7% (1)
*p*	1.000	-	1.000			

A comparison was made between the same part of lamellae. The number of lamellae is shown in brackets. Significant differences (*p* < 0.05) are bolded.

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
