# Peer review of "The Differences in Histoarchitecture of Hoof Lamellae between Obese and Lean Draft Horses"

_animals, 2022, doi:10.3390/ani12141774_

Round 1
Reviewer 1 Report
Thank you for this interesting paper looking at the influence of obesity on lamellar failure in draft horses by comparing forelimbs of 6 obese and 6 lean horses. I think the authors would do themselves a big favour by explaining a little bit more in the introduction as to why draft horses were chosen for this study as well as in the discussion how the management of what I understand to be meat horses may differ from those kept for leisure or sport. Further, there are some considerable gaps in the methods section that need to be filled in order to assist the reader in fully understanding what was done and how, and help with interpretation of the results.
Simple summary
Line 13 – obesity is associated with EMS rather than correlated
Line 15 – exclude rather than reject
Line 17 – differed rather than differs
Line 17 – how did lamellae differ and what does damage mean? You say that there were differences between lean and obese animals but then say that there was no effect of obesity on lamellar failure. So how was this different determined – they type of changes were different rather than the number of changes?
Line 18 – this study rather than the research
Line 19 – support rather than confirm
Here and throughout you use quite a number of different terms to refer to the lamellae, including laminae, lamellar and laminal. I would strongly suggest you pick one term and use the corresponding terms throughout e.g. I would recommend lamellae and lamellar
Abstract
Line 23 – association rather than correlation
Line 26 – exclude rather than rejected. Radiographs were used to exclude laminitis – specifically to exclude chronic laminitic changes because subtle sub-clinical or acute-phase changes would unlikely be visible on a radiograph
Introduction
Line 43 – The key characteristic of EMS is insulin dysregulation (ID) rather than insulin resistance, which may not always be present but can develop as a consequence of prolonged hyperinsulinaemia which then feeds into and perpetuates the cycle. See the ECEIM consensus statement on equine metabolic syndrome for a good explanation: https://www.ncbi.nlm.nih.gov/pmc/articles/PMC6430910/
Line 44 – how do you define primitive breeds?
Line 46 – do you have a reference for the statement that EMS affects mature horses? Although it has been found that ID does tend to increase with increasing age, I know in a recent UK study that investigated prevalence and risk factors for EMS in native ponies and cobs, the median age of ponies that were found to have EMS was 11 years (IQR 9-13) although a range is not stated (see Carslake et al https://beva.onlinelibrary.wiley.com/doi/full/10.1111/evj.13378). Also, usually EMS is often only diagnosed after the horse has had one or several active episodes of laminitis so its onset may be much earlier before it is actually diagnosed.
Line 46 - EMS is not a disease in itself but rather a collection of characteristics or risk factors which predispose horses to endocrinopathic laminitis and can also occur in lean animals in some cases despite it’s strong association with obesity (see consensus statement above).
Line 47 – understanding insulin dysregulation
Line 53 – pathophysiology rather than pathomechanism
Line 55 – endocrinopathic rather than metabolic issues
Line 56 – Endocrinopathic rather than metabolic
Line 70 – ID instead of IR
Line 71 – delete the before obesity
Line 72 – you have not included any explanation or reasoning for focusing on draft horses in your introduction – why draft horses in particular? This needs to be added.
Materials and Methods
Line 75 - how was the sample size of 12 derived? Were both forelegs from each horse collected or just one. If one, how was left or right chosen?
Line 76 – who performed the body condition scoring?
Line 81 – How was insulin activity assessed? I am guessing you tested basal serum insulin concentrations? What was the cut-off which determined whether the horse had ID and thus EMS and would thus be excluded? You don’t mention any exclusion criteria here although you include them in the abstract – these need to be explained.
Line 100 – please include measurement units for PELL, KPELL and TELL
Line 109 – Please can you explain if insulin concentrations were normally distributed (I assume they are as you seem to report means and standard deviations in the results section [although this is unclear] and you have used a Student’s t-test) and how normality was assessed?
Line 109 – same question about normality for PELL, KPELL and TELL measurements as above. Also I am a little bit confused as to how the mean of these measurements were determined as from my understanding several measurements were taken per foot/per horse – were these then combined to get a mean per horse (which might also identify some individual variation between horses) or were all measurements in lean horses and all measurements in obese horses combined together (irrespective of whether they belonged to same/different horses) to give a group mean? This needs to be explained.
Line 111 – what were the different shape classifications/categories? – I know you’ve referenced Karikoski et al, 2009 but it would be useful to briefly explain this classification.
Results
Line 114 – are these means and standard deviations? If so, please state this under statistical methods.
Line 115 – please report the p-value result even if it was not statistically significant.
Line 115 – so was 20 µg the cut-off used to determine whether to exclude a horse? Thus no horses were diagnosed with basal hyperinsulinaemia and therefore excluded?
Line 116 – therefore none were excluded based on having chronic laminitic changes?
Lines 117-121 – Did you visualise the range of individual variation between horses? This would be interesting to present/comment on.
Line 120 – remove Surprisingly. Start with “There were no differences…”
Line 121 – what was the exact p-value?
Line 126 - please report the p-value result even if it was not statistically significant.
Tables 1, 2 and 3 – commas should be replaced with decimal points. Also can you include the actual number of observations within each group along with the percentage? I have several other questions which relate to the tables and statistical methods:
- Were separate Chi-squared test performed for each PEL and PDL shape between obese and lean groups
– If yes the it would be worth adding another column and reporting the actual p-value for each comparison rather than just saying p<0.05 and p<0.001)?
- If so, were results adjusted for multiple comparisons?
- Some of your groups have zero observations or may have small numbers (hard to tell without seeing the actual number in each group rather than just percentage) but in that case you should be using a Fisher’s exact test rather than a Chi-squared test.
Discussion
Line 180 – here and throughout check spelling of endocrinopathic; experimentally induced
Line 220 – This sentence needs to be rewritten as it currently does not make sense. Further to horses being kept for meat – I think you need to elaborate some more on how their management would differ from leisure/sport horses – are they purposefully fed extra calories to fatten them up? Were any of them assessed for lameness prior to slaughter? Do they receive little/no exercise? What could help explain what you are seeing in the lamellae that is different to other reported studies other than breed?
Line 224 – I think several studies have now shown that basal serum insulin concentrations are a good a predictor of laminitis risk (and as such, ID) so while I agree that dynamic testing would have been ideal, I don’t feel that this is a great limitation.
Line 230 – This isn’t a conclusion that has come about as a direct finding from your study i.e. you did not examine warmblood lamella and statistically compare the length with draft lamellae. So perhaps something like “The findings of from this study suggest breed differences in the morphology and length of lamellae may exist with draft horses appearing to have longer PEL than previously reported in other breeds.”
What conclusions can you reach from the differences in shape you did identify?
Author Response
Dear Reviewer #1,
We would like to thank you very much for Your review and important remarks that helped to improve the quality of our manuscript. All the comments were carefully considered and the detailed responses are provided in attached file.

Reviewer 2 Report
Dear authors,
Many thanks for this piece of work. I found merit in it the idea behind the manuscript, though in my opinion there is a background flaw.
Your manuscript reports an observational study and not an experimental study in which influences can be described after a period of experimental treatment. What you can do is a correlation (as the best) between the fatness vs leanness of horses and lamellar histoarchitecture or arrangement. Nothing more. I also find inappropriate the determination of insulin at non fasting time and in one only shot sample, without a description of previous diets and moreover after (I suppose) transportation and stressing of horses prior to slaughtering.
In my opinion the manuscript should be turned into a short communication in which a correlation on a semi quantitative appearance of the tissue (develop a scoring system) can be graded with fatness of horses.
Finally, when you state a BCS 9/9 (I think according to Henneke et al. scoring system) you should not report extremely obese. It is obese. Just use extremely overweight, gaining a 9/9 (obese).
Lastly, I am not totally aware of the obesity as a common problem in horses (all horses??? Breeds and types????? Are you sure????). You cite one (quietly dating back in time) reference to state this overall assumption.
I don't think the paper should be rejected, but surely it needs to be tailored to the results because it overstated (apparently) what was actually done. Unless, it was not adequately described and I am wrong.
Author Response
Dear Reviewer #2,
We would like to thank you for Your review and important remarks that helped to improve the quality of our manuscript. All the comments were carefully considered and the detailed responses are provided in attached file. We have changed a lot in our paper and we humbly hope you would like it more.

Round 2
Reviewer 1 Report
Thank you to the authors for making the suggested changes to help improve their paper. Please find some additional comments below that should be addressed before this paper is published.
Title – change laminae to lamellae (as I think this is now what you use throughout the body of the paper)
Simple summary
Line 16 – spelling of exclude
Line 20 – primary dermal lamella (PDL)
Line 20 – proportion or percentage rather than rate
Line 24 – PEL should be in brackets (PEL)
Abstract
Line 29 – exclude rather than excluded
Line 55 – insulin dysregulation rather than resistance
Introduction
Line 52 – you need to define what IQR is – since I don’t think you use it again you probably only need to say (with an interquartile range between 9 and 13).
Line 52 – as said previously EMS/ID are not a disease but a syndrome. Also it is usually diagnosed later in a horse’s life but may be present already for many years. So I would rephrase the start of this sentence to say “Insulin dysregulation is usually diagnosed in adult horses over 11 years of age…”
Materials and Methods
Line 88 – left missing t
Line 93 – obtain rather than perform
Line 97 – insulin has an extra e at the end
Line 97 – please confirm the units of measurement for the insulin
Line 98 – you need to include a sentence saying that radiographs were taken (how many, from which direction, etc.) to exclude any changes associated with chronic laminitis and you should also say what range of distal phalanx rotation was deemed acceptable (in results you mention all horses had <4 degree rotation). It is very important in the materials and methods to clearly state and define everything that was done.
Line 132 – If the number of comparing groups is <5 then the Fisher’s Exact test would be used
Results
When reporting p-values it is normal convention to include up to three numbers after the decimal point e.g. p=0.9221 should be p=0.922. Where you have very small p-values e.g. p=0.000002 then it is OK to use p<0.001. Please also make sure all commas are replaced with decimal points.
Table 3 – check your bolding as some non-significant comparisons also seem to be bolded
Regarding my question asking if results were adjusted for multiple comparisons using something like the Bonferroni adjustment or similar. I’m guessing not, but you need to add this to the statistical methods and say no adjustment were made and why, ideally quoting a reference. There are arguments for and against using multiple comparisons so you just need to justify your choices.
Discussion
Line 245 – I’m not quite sure I understand this sentence. So are you suggesting that the non-obese horses could still have had sub-clinical laminitis and associated lamellar changes but without gaining weight? What does “the disease” refer to? I think this sentence should be re-written to make your point more clear.
Author Response
Dear Reviewer,
We analyzed your points carefully and answered for all. We attached the file with our responses. We truly appreciate your review of our work. We are grateful for all the tips, that helped us get the best out of our research. We hope that the paper already meets your expectations.

Reviewer 2 Report
Dear authors,
Thank you for welcoming my suggestion. You are right. I like you ms much more than before. However, now, what is left to improve is the English grammar and style.
Thank you.
Author Response
Dear Reviewer#2
We are really glad you liked our research more. We improved the grammar and style. We hope our paper is meeting your expectations.
Thank you,